# A Rearrangement of the Mitochondrial Genes of Centipedes (Arthropoda, Myriapoda) with a Phylogenetic Analysis

**DOI:** 10.3390/genes13101787

**Published:** 2022-10-03

**Authors:** Jia-Jia Wang, Yu Bai, Yan Dong

**Affiliations:** College of Biology and Food Engineering, Chuzhou University, Chuzhou 239000, China

**Keywords:** centipedes, *Mecistocephalus marmoratus*, mitogenome, phylogenetic relationship, rearrangement, *Scolopendra subspinipes*

## Abstract

Due to the limitations of taxon sampling and differences in results from the available data, the phylogenetic relationships of the Myriapoda remain contentious. Therefore, we try to reconstruct and analyze the phylogenetic relationships within the Myriapoda by examining mitochondrial genomes (the mitogenome). In this study, typical circular mitogenomes of *Mecistocephalus marmoratus* and *Scolopendra subspinipes* were sequenced by Sanger sequencing; they were 15,279 bp and 14,637 bp in length, respectively, and a control region and 37 typical mitochondrial genes were annotated in the sequences. The results showed that all 13 PCGs started with ATN codons and ended with TAR codons or a single T; what is interesting is that the gene orders of *M. marmoratus* have been extensively rearranged compared with most Myriapoda. Thus, we propose a simple duplication/loss model to explain the extensively rearranged genes of *M. marmoratus*, hoping to provide insights into mitogenome rearrangement events in Myriapoda. In addition, our mitogenomic phylogenetic analyses showed that the main myriapod groups are monophyletic and supported the combination of the Pauropoda and Diplopoda to form the Dignatha. Within the Chilopoda, we suggest that Scutigeromorpha is a sister group to the Lithobiomorpha, Geophilomorpha, and Scolopendromorpha. We also identified a close relationship between the Lithobiomorpha and Geophilomorpha. The results also indicate that the mitogenome can be used as an effective mechanism to understand the phylogenetic relationships within Myriapoda.

## 1. Introduction

Centipedes, also known as the Chilopoda (CHI), and their related groups (Diplopoda (DIP), Symphyla (SYM), and Pauropda (PAU)) comprise the subphylum Myriapoda. Most centipedes are fast-moving, have a predatory lifestyle in terrestrial habitats, and possess poisonous modified maxillipeds [1]. Centipedes comprise more than 3000 species in five extant orders: Scutigeromorpha, Lithobiomorpha, Craterostigmomorpha, Scolopendromorpha, and Geophilomorpha [2,3]. In some studies, *Strigamia maritima* is treated as an ideal model species for ecological and developmental research [4,5]. Recent studies using either comparative morphological or molecular evidence have found that myriapods and all extant myriapod classes are monophyletic [6,7,8,9,10,11,12,13,14,15,16,17,18,19,20]. Most of these analyses support the contention that the Chilopoda represent the basal lineage of the Myriapoda, with the remaining three classes united as the Progoneata.

However, the phylogenetic relationships among the major extant groups of myriapods remain uncertain, and recent debates have focused on the Edafopoda hypothesis (PAU + SYM group) vs. the Dignatha hypothesis (PAU + DIP group). The Dignatha hypothesis has been universally accepted for more than a century [8,10,20,21,22,23,24,25,26], whereas that of Edafopoda is corroborated by nuclear ribosomal genes and some mitochondrial genomes (mitogenomes) [12,14,17]. However, recent phylogenomic analyses based on transcriptomic data showed slightly different results. The results obtained by Szucsich et al., Benavides et al., and Wang et al. are basically the same, demonstrating strong evidence for the clade Pauropoda + Symphyla (=Edafopoda) as well as for Chilopoda + Diplopoda (=Pectinopoda) [19,21,22]. The difference is that Fernández et al. identified two alternative phylogenetic relationships for Symphyla: one that classifies it as a sister group to the Diplopoda + Chilopoda, and one that places Symphyla closer to Dignatha [27].

The relationships between Chilopoda clades have been elucidated using morphological characteristics and molecular analyses, as follows: (1) the five centipede orders are all monophyletic; (2) the basal division, Scutigeromorpha, is a sister group to the four other centipede lineages; and (3) the Scolopendromorpha + Geophilomorpha comprise the clade Epimorpha [11,27,28,29,30]. However, some transcriptome-based phylogenetic analyses did not reveal a sister group relationship between Geophilomorpha and Scolopendromorpha [24], whereas others did [31]. Thus, centipede phylogeny remains a topic that needs further investigation in terms of the position of different centipede lineages within the chilopod orders and the earliest evolutionary splitting events within the centipede lineages.

Arthropod mitogenomes encode 13 protein-coding genes (PCGs), 22 transfer RNA genes (tRNAs), two ribosomal RNA genes (rRNAs), and a long noncoding region (control region, CR), which have been extensively used to study genetics and evolution at multiple hierarchical levels [31,32,33,34,35]. The order of mitochondrial genes can provide additional phylogenetic information because mitochondrial gene rearrangements are generally rare events. Moreover, most mitochondrial gene arrangements are generally stable in arthropods over a long evolutionary period. Wang et al. found two types of gene orders in the Neuropterida [35]. One was the same as the ancestral mitochondrial gene orders of most insects, and the other was the result of a shuffle of *trnC* to upstream of *trnW* (*trnC-trnW-trnY*), an arrangement that is present in all remaining families of Neuroptera. Song et al. reported that various rearrangements of the hotspot region between the CR and *cox1* are found in several insects; all species of the Lepidoptera suborder Ditrysia had the arrangement *trnM-trnI-trnQ*, and most species of Neuroptera had the transposition of *trnW* and *trnC* [34]. Most hymenopterans have the *trnI*, *trnQ*, and/or *trnM* genes in different positions; for example, the *trnM-trnI-trnQ* order is the most common in Formicidae [36,37].

Although taxon sampling has been limited, the mitogenome has provided evolutionary evidence related to the phylogenetic and evolutionary histories of the Myriapoda. At the same time, a variety of gene rearrangement events have been found. The gene orders in the mitogenomes of *Cermatobius longicornis* (centipede) and *Prionobelum* sp. (millipede) are identical to those in *Limulus polyphemus* (Arthropoda: Xiphosura) [38,39,40,41]. All the sequenced mitogenomes of millipedes have a *nad6* + *cob* placement that differs from that of *L. polyphemus*, except Sphaerotheriida; the pattern of *nad6* + *cob* is believed to be reliable molecular evidence supporting the Helminthomorpha clade, and the inversion of the entire side of a genome (the *trnF-nad5-trnH-nad4-nad4L* cluster, *trnP*, the *nad1-trnL2-trnL1-rrnL-trnV-rrnS* cluster, *trnQ*, *trnC*, and *trnY*) could represent a synapomorphy of a subgroup within Polydesmida [13]. Nine myriapod mitogenomes were compared by Dong et al., who posited that a translocation of *trnT* from the 5′ end of *nad4L* was a common event in derived progoneate lineages [12]. Xu et al. sequenced the first Spirobolus mitogenomes and analyzed the phylogenetic relationships within Diplopoda based on 9 orders and 27 species [42].

In previous studies, the mitochondrial gene orders of *Scolopendra dehaani*, *Scolopendra mutilans*, *S. maritima*, *Scutigera coleoptrata,* and *Spirobolus bungii* were distinctly different from those of any other myriapod species [42,43,44,45]. A high rate of rearrangement makes the Myriapoda an ideal class group for exploring the interactions between gene rearrangements and phylogenetic relationships. Further sequencing of mitogenomes from additional members of the Chilopoda can demonstrate whether such an extensive rearrangement is unique. Common models that attempt to explain gene rearrangement events and investigate the evolutionary implications of these events involve duplication–random loss (TDRL) and duplication–nonrandom loss (TDNL) as the molecular drivers of gene rearrangement [32].

In this study, complete mitogenomes of *M. marmoratus* and *S. subspinipes* were sequenced and annotated, and we used mitogenomes to investigate the gene rearrangement model and the phylogenetic relationships within centipedes, hoping to provide more molecular evidence to explore the relationships within the Myriapoda.

## 2. Materials and Methods

### 2.1. Taxon Sampling and Mitochondrial DNA Sequencing

Specimens of *M. marmoratus* and *S. subspinipes* were collected from the Langya mountains, Chuzhou, Anhui province, China (32°16′ N, 118°16′ E), in July 2014. They were initially preserved in 100% ethanol in the field and were then transferred to −20 °C conditions for long-term storage at the Molecular Biology Laboratory of Chuzhou University (Chuzhou, China). Genomic DNA was extracted from the dehydrated muscle tissues using DNeasy Blood and Tissue kits (Qiagen, Hilden, Germany). The entire mitogenome was amplified using six primer pairs (Table 1), and all the primers were valid reference primers that had been used for the published species. After using and screening, the six most suitable primer pairs were obtained. Short polymerase chain reaction (PCR) assays (<1.5 kb) were performed using *KOD* Dash DNA polymerase (Toyobo). The cycling conditions were 94 °C for 5 min; followed by 35 cycles of 30 s at 94 °C, 50 s at 49–52 °C, and 1–2 min at 72 °C (depending on the amplicon size); with a final elongation step at 72 °C for 10 min. Long PCR assays (>1.5 kb) were performed using LA *Taq* DNA polymerase (TaKaRa). The two-step conditions were as follows: 35 cycles at 96 °C for 2 min and 68 °C for 10 min, followed by incubation at 68 °C for 10 min. The amplified PCR products were electrophoresed on 2% agarose gel, excised, purified, and then analyzed by primer walking on an ABI-PRISM 3730 Automated DNA Sequencer (Applied Biosystems, Waltham, MA, USA).

### 2.2. Gene Annotation and Secondary Structure Prediction

PCR product sequences were assembled using SeqMan II (DNASTAR Inc., Madison, WI, USA) after checking. Preliminary annotation using the MITOS web server (http://mitos.bioinf.uni-leipzig.de/index.py (accessed on: 10 October 2021)) provided overall information on the mitogenomes [38]. Further annotation of 13 PCGs was performed by identifying their open reading frames and aligning them with homologous genes from other reported myriapod mitogenomes from the NCBI database (https://blast.ncbi.nlm.nih.gov/Blast.cgi (accessed on: 11 October 2021)). tRNA genes were identified by comparing the results predicted using the software programs tRNAscan-SE Search Server v2.0 and ARWEN [48,49]. Based on known gene order information, the boundaries of *rrnL* (*16S* rRNA) were assumed to be delimited by the ends of the *trnV* and *trnL2* pair. Further, *rrnS* (*12S* rRNA) was assumed to start from the end of *trnV*, and its end was roughly identified by alignment with the other published millipede sequences. Nucleotide frequencies and codon usage were determined using MEGA X software [50].

### 2.3. Sequence Alignment and Phylogenetic Analyses

Phylogenetic trees were constructed based on 32 ingroups and 2 outgroups (Appendix A), and *Priapulus caudatus* and *Epiperipatus biolleyi* (GenBank accession numbers NC_008557 and NC_009082, respectively) were selected as the outgroups in our analyses. Datasets of 2 rRNA sequences and 13 PCG sequences were selected to analyze the phylogenetic relationships within Myriapoda. The sequences of PCG genes were initially aligned using MASCE v2, and rRNA genes were initially aligned using MAFFT with the E-INS-I strategy [51,52]. Poorly aligned positions were subsequently eliminated using Gblock 9.1b, with default settings [53]. Finally, we used MEGA to concatenate all genes and selected their positions to form two datasets: (1) the PCG12RNA matrix and the first and second codon positions of the 13 PCGs and 2 rRNAs, for a total of 7954 bp; and (2) the AA matrix and the amino acid sequences of 13 PCGs, for a total of 3451 aa.

The optimal partition scheme for each dataset and the best model for each partition were determined using Partition Finder 2 (Appendix A), with the Akaike information criterion model and a greedy search algorithm with unlinked branch lengths [54]. We analyzed the phylogenetic relationships using the BI and ML methods using the IQ-TREE and MrBayes v3.2.6 under models, respectively [55,56]. For ML analyses, we used an ultrafast bootstrap approximation approach with 1 × 10^4^ replicates, whereas for BI analyses, we used the default settings by simulating four independent runs for 1 × 10^7^–5 × 10^7^ generations and sampling every 100 generations after the average standard deviation of split frequencies fell below 0.001. The first 2000 trees were discarded as burn-in. Three replicates of these BI runs were conducted, all of which produced the same topology.

## 3. Results and Discussion

### 3.1. Organization of the Mitogenome

As shown in Figure 1 and Appendix A, the complete mitogenomes of *M. marmoratus* (KX774322) and *S. subspinipes* (MN642577) were sequenced and annotated. The length of the *M. marmoratus* mitogenome was 15,279 bp, and that of *S. subspinipes* was 14,637 bp. Both complete mitogenomes included 37 typical mitochondrial genes—two rRNA genes (*rrnS* (*16S* rRNA) and *rrnL* (*12S* rRNA)), 13 PCGs (*cox1-3*, *cob*, *nad1-6*, *nad4L*, *atp6*, and *atp8*), 22 tRNA genes, and a control region (CR) (Appendix A). The sizes of these two mitogenomes were within the range reported for known myriapod mitogenomes, from 14,487 bp (*Pauropus longiramus*) to 16,833 bp (*C. longicornis*) [13]. The A + T content in the mitogenomes of Chilopoda ranged from 63.4% (*C. longicornis*) to 78.8% (*S. mutilans*). The A + T contents in the mitogenomes of *M. marmoratus* and *S. subspinipes* were 69.5% and 72.7%, respectively. Additionally, both genomes showed an obvious A + T and C + G bias (Table 2).

The length variation was minimal in the PCGs, with greater variation in the putative CR, intergenic overlaps, and tRNAs. Frequent intergenic overlaps (17/37 = 46%) occurred in the mitogenome of *S. subspinipes*. The two newly sequenced mitogenomes contained the 37 genes commonly found in metazoan mitogenomes as well as a putative CR, including the presumed origin of replication and promoters for transcription initiation. All PCGs started with ATN. The AT contents of the PCGs of Chilopoda ranged from 59.6% (*S. dehaani*) [43] to 77.2% (*S. mutilans*) [41]. The A + T contents of the PCGs in the mitogenomes of *M. marmoratus* and *S. subspinipes* were 67.1% and 71.6%, respectively. Additionally, both genomes showed an obvious T + A bias and a slight C + G bias, except *S. dehaani*, which has an obvious A + T bias, and *S. mutilans*, which has a slight G + C bias (Table 2).

### 3.2. Transfer RNAs

The secondary structures of the 22 potential tRNA genes in *M. marmoratus* and *S. subspinipes* were predicted and are shown in Appendix A, respectively. The newly sequenced mitogenome of *M. marmoratus* revealed the loss of the dihydrouridine arm in *trnC* and *trnS1*. Among the 22 tRNAs in the mitogenome of *S. subspinipes*, *trnT* and *trnP* lacked a TΨC loop, and *trnS1* lacked the DHU arm. Many tRNA genes in the newly sequenced mitogenomes were shortened, with the shortest tRNAs having only 52 nucleotides. A total of 22 tRNAs ranged in length from 52 bp (*trnS1*) to 79 bp (*trnW*) in *S. subspinipes*, and from 52 bp (*trnT*) to 73 bp (*trnN*) in *M. marmoratus*. This difference was mainly caused by the loop region, particularly the variable loop.

### 3.3. Phylogenetic Analyses

As in the previous study, the monophyly of the Myriapoda and the three classes (CHI, DIP, and SYM) were verified using phylogenomic analyses [12,13,19,21,22]. This is considered to be uncontroversial; the current controversy is the phylogenetic relationships among these groups, including PAU. However, the phylogenetic relationships among centipedes reported in previous studies differ due to differences in the molecular markers and analysis methods used [16,27,31,43,44]. Our study provided compelling support for Dignatha (DIP + PAU) being the closest relatives, which supported the relationships among the four groups that were suggested based on morphological and some molecular evidence [24], but was in conflict with the results of phylogenomic analyses with transcriptomic data published by Szucsich et al. and Benavides et al. [19,21]. Regrettably, the position of SYM in the phylogenetic tree may be unstable due to the limited number of SYM samples used in this study. In our study, we also observed that different results were obtained from different datasets. In the trees produced using the AA matrix (Figure 2a), Chilopoda was identified as the basal lineage of the remaining myriapods and as a sister group with the Progoneata, in both the BI and ML analyses. The relationship within Progoneata was established as (DIP + PAU) + SYM; this result is consistent with earlier results published by Fernández, Edgecombe, and Giribet and is further consistent with the morphological evidence [24]. The results produced using the PCG12RNA matrix were slightly different (Figure 2b). The topology DIP + PAU, with a sister group of SYM + CHI, was identified with high support based on the BI and ML analyses. Thus, in order to clarify the phylogenetic relationships among the major groups of Myriapoda, we need to develop new methods and increase the sampling richness.

Focusing on the Chilopoda, *M. marmoratus* and *S. maritima* grouped into one branch, which corresponds to the Geophilomorpha in all trees. In the Scolopendromorpha, the relationships between *S. mutilans*, *S. dehaani*, and *S. morsitans* were closer than the relationship of any of them to *S. subspinipes*. Contrary to the phylogenetic relationship, the gene orders of the mitogenomes of *S. dehaani* and *S. subspinipes* were closer; *trnE* is missing in *S. dehaani*. Thus, to verify the relationship between evolution and gene order within the Scolopendromorpha, we need to systematically add more samples to identify the rules of gene rearrangement and clarify the relationships between genera. In previous studies, Scutigeromorpha was identified as a sister group to three other groups, namely Lithobiomorpha, Geophilomorpha, and Scolopendromorpha [19,27,31]. We found that the relationship between the Lithobiomorpha and Geophilomorpha was closer, although traditional morphological evidence indicates that Geophilomorpha and Scolopendromorpha are more closely related. These relationships identified in the present study are consistent with the conclusions obtained from a previous study [43].

Taxon sampling is essential for the accuracy of phylogenetic inference, and it is important to understand the effect of taxon sampling in a whole-genome or multi-locus phylogenetic study [57]. More Pauropoda should be included, and the selection of different markers will be necessary in future studies to reconstruct a stable phylogenetic topology.

### 3.4. Evolution of Gene Rearrangements in the Mitochondrial Genome of Centipedes

Previous studies have shown that the mitogenomes of members of the class Chilopoda are characterized by extensive mitochondrial gene rearrangements [40,41,43,44,45,58]. In the newly sequenced mitogenomes, the gene arrangement in *S. subspinipes* was identical to that in *L. polyphemus*. Within the Scolopendra, *S. dehaani* and *S. subspinipes* have the same gene order, except that *S. dehaani* does not possess *trnE* and *trnL2*. Compared with these two species, five genes or gene blocks in the mitogenome of *S. mutilans* (*trnF-nad4L*, *trnP-trnS2*, *nad1*-CR, *trnV*, and *trnQ*) were rearranged. Compared with ancestral gene orders of myriapods (*L. polyphemus*), at least eight genes and gene clusters (*nad3*, *nad6-cob*, *trnM-nad2-trnW*, *trnT*, *trnN*, *trnY*, *trnL1*, and *trnI*) in the mitogenome of *M. marmoratus* were rearranged. The overall arrangement of genes in the mitogenome of *M. marmoratus* was unique compared with that of the mitogenomes of other myriapod species or any other arthropods that have been studied. The same transcriptional polarity genes were clustered together, except *trnS2*, which overlapped with the genes on the opposite strand.

In order to better understand the evolution of the extensive mitochondrial gene rearrangements of centipedes, the gene orders of published centipede mitogenomes were summarized and mapped on the estimated phylogenetic tree (Figure 3). The gene arrangement of the Lithobiomorpha is relatively conservative, and other classes have been extensively rearranged. At present, the most widely accepted model involves duplication–random loss (TDRL) and duplication–nonrandom loss (TDNL) to explain the molecular drivers of gene rearrangement [59]. A recent study on the pattern of gene rearrangements in *Polydesmus* sp. GZCS-2019 suggested a new rearrangement model based on three factors: genome-scale duplication, loss, and recombination (TD(N/R) L + C) [21]. This model provides a reference for improving the understanding of the mechanism of gene rearrangement in Myriapoda. However, several unique rearrangement units of *M. marmoratus* prevent the application of these models to the species.

In this analysis, we assumed that the original mitogenomes of *L. polyphemus* had an identical gene arrangement, which appears to be ancestral for myriapods and arthropods (Figure 4A). We propose an immature duplication/loss (random and nonrandom) model that resulted in the generation of the mitochondrial gene arrangement of *M. marmoratus* (Figure 4).

Our hypothesis involves three steps. First, two derivative monomers are arranged in a circular dimer, similar to that found in some myriapods, for example, *Narceus annularus*, *Thyropygus* sp., *Antrokoreana gracilipes*, *Trigoniulus corallines*, *Abacion magnum*, *Brachycybe lecontii*, and *Symphylella* sp. Subsequently, nonrandom loss occurred according to the transcriptional orientation of each gene. Second, duplication and nonrandom loss events are necessary to explain the translocation of *nad3*, *trnF-nad5-trnH-nad4-nad4L-trnT-trnP*, and *trnQ-trnM-nad2-trnW* (Figure 4B1). In this study, the model of duplication and random loss of genes was used to explain the translocation of the mitochondrial genes *trnN*, *trnL1*, and *trnI* (Figure 4B2) [39]. Thus, the mitochondrial gene arrangement in *M. marmoratus* was deduced from the original gene order (Figure 4C). Gene density in a duplicated region can be determined via biological constraints rather than by chance. Because mitochondrial gene rearrangements are rare events in animal evolution, they appear to be well-suited for deriving phylogenetic inferences from ancient relationships.

The highly unusual organization of the mitogenome of *S. maritima* is possibly due to the stem and loop structures. The rare gene cluster with opposite transcriptional polarity in the mitogenome of *M. marmoratus* suggests that a nonrandom mechanism is involved in generating this gene order. Based on these two sequenced mitogenomes (*S. maritima* and *M. marmoratus*) within Geophilomorpha, gene orders are derived using different mechanisms. In order to explain the reason for the unusual organization of each mitogenome within Geophilomorpha, more mitogenomes need to be sequenced in the future.

The gene transfer and gene block arrangements may represent a synapomorphy in the related lineage, resolving the phylogenetic controversy at multiple hierarchical levels [13]. We believe that meaningful evolutionary information can be obtained by comparing the gene order of myriapod species, provided that data on broader taxon sampling are available. Ultimately, the use of a large number of samples would help elucidate the evolutionary details.

## 4. Conclusions

At present, we report the complete mitogenomes of *M. marmoratus* (15,279 bp) and *S. maritime* (14,637 bp) (Myriapoda: Chilopoda). Both mitogenomes contain 37 typical genes, and the gene order of *S. subspinipes* was the same as that of the original arthropod and myriapod mitogenome (*L. polyphemus*), whereas *M. marmoratus* changed considerably. A simple duplication/loss (random and nonrandom) model was proposed to explain the mitochondrial gene arrangement in *M. marmoratus*; we hope that this model can also reveal mitochondrial gene rearrangement events in other species. Further, we explored the phylogeny of Myriapoda based on mitogenomes. The results suggested the monophyly of the Myriapoda and its main groups and supported Pauropoda and Diplopoda forming the Dignatha, although the relationship between Dignatha and Symphyla needs to be explored further with more systematic evidence. Within the Chilopoda, Scutigeromorpha was a sister group to three other groups: Lithobiomorpha, Geophilomorpha, and Scolopendromorpha. Our study identified a close relationship between Lithobiomorpha and Geophilomorpha. Overall, our results help to better understand the phylogeny of the Myriapoda and explain gene rearrangement events.

## Figures and Tables

**Figure 1 genes-13-01787-f001:**
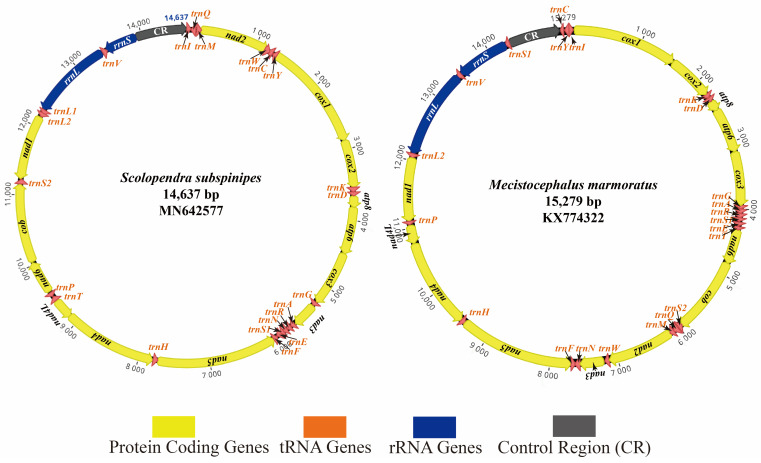
Circular map of the mitogenomes of *S. subspinipes* and *M. marmoratus*.

**Figure 2 genes-13-01787-f002:**
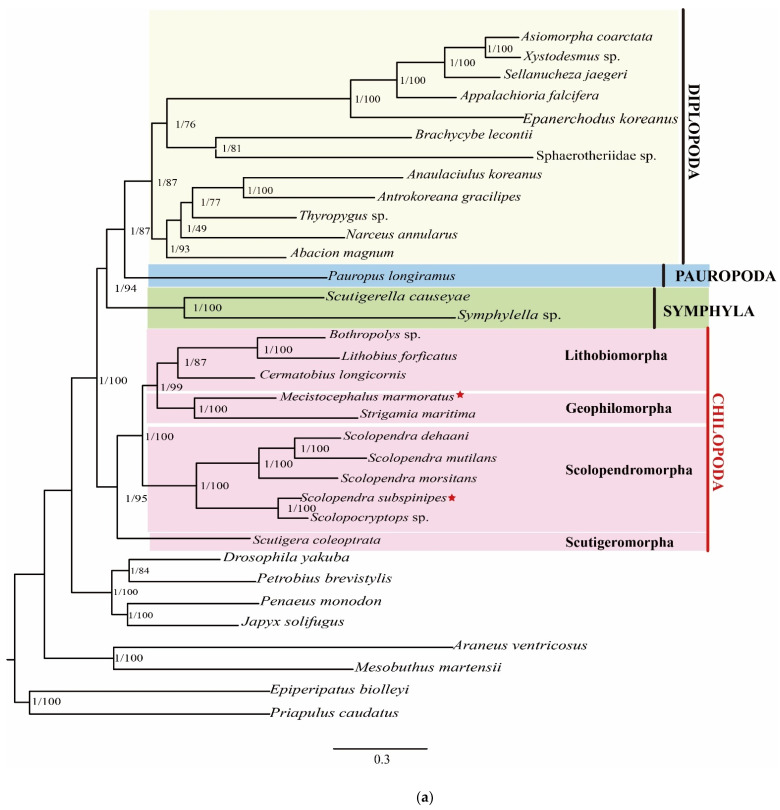
(**a**) Phylogenetic trees of myriapods inferred using Bayesian inference (BI) and the maximum likelihood method (ML) based on the AA matrix. Numbers on the branches indicate the posterior probability (BI) (**left**) and bootstrap values (ML) (**right**). Newly sequenced taxa are indicated using red stars. (**b**) Phylogenetic trees of myriapods inferred using the MrBayes program with the maximum likelihood method based on the PCG12RNA matrix. Numbers on the branches indicate the posterior probability (BI) (**left**) and bootstrap values (ML) (**right**). Newly sequenced taxa are indicated using red stars.

**Figure 3 genes-13-01787-f003:**
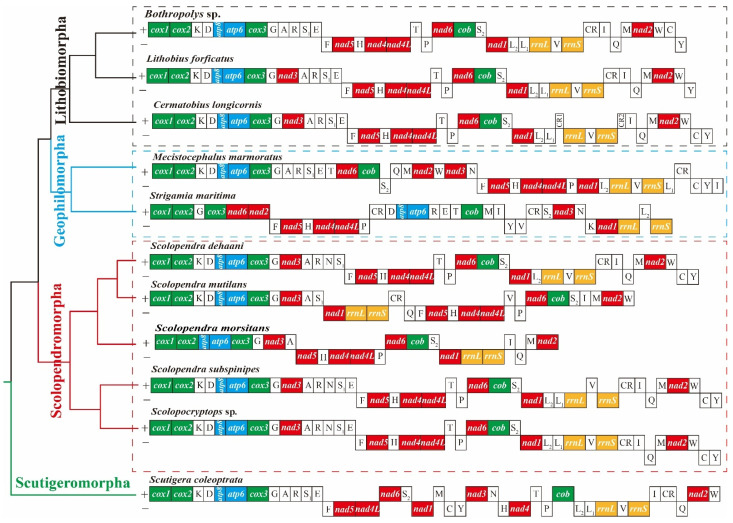
Gene arrangements of the Chilopoda mitochondrial genomes. The mitogenomes have been linearized for ease of comparison and arbitrarily begin with cox1 when possible. Different genes are shown in different colors. Underlined labels indicate that the gene was transcribed from the minority strand.

**Figure 4 genes-13-01787-f004:**
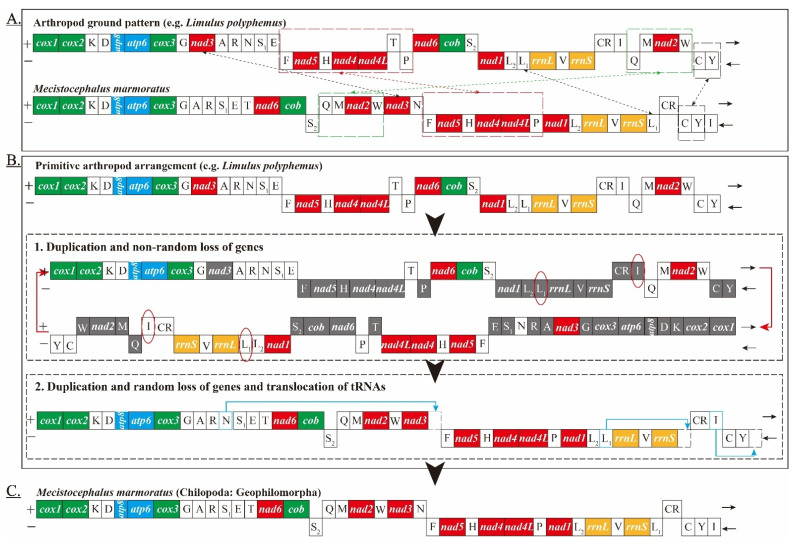
Inferred intermediate steps between *Limulus polyphemus* and *M. marmoratus*. The lost genes are labeled in gray. (**A**) The ancestral gene arrangement of the myriapod. (**B1**) Two monomers derived from the duplication of the ancestor arranged in a circular dimer. Subsequently, nonrandom loss occurred according to the orientation of transcription for each gene. (**B2**) Tandem duplication followed by the random loss of genes and the translocation of tRNAs. (**C**) Final gene orders of the *M. marmoratus* mitogenome.

**Table 1 genes-13-01787-t001:** PCR primers used in this study.

Primer Name	Nucleotide Sequence (5′-3′)	PCR Amplification Product Length	Reference
CO1CF	GCACGTCTACAAATCATAAAGATATTGG	0.7 kb	[46]
CO1CR	TAAACTTCAGGGTGACCGAAAAATCA		
Lco1	TTATAATTTTTTTTATAGTGATACC	3.7 kb	[12]
CO3R	ACATCTACAAAATGTCAGTATCA		[47]
Dco3F	TATCATCCTATCAATGATGACGAGA	3.7 kb	[12]
Dn4R	ATTTATGATTACCTAAGGCTCATG		
Dn4F	ATGAACAACAGAAGAATAAGC	2.9 kb	
Hcob	GCAAATAAAAAATATCATTCTGGTTG		
DcobF	ATAATAACCGCCTTCTTGGGAT	3.4 kb	
D12SR	CTGTTTCTGAATCGATATTCCACGTTT		
D12SF	ATAATAGGGTATCTAATCCTAGTCT	2.7 kb	
Dco1R	ATGGGGGATATACGGTCCATCCGG		

**Table 2 genes-13-01787-t002:** Nucleotide composition of the mitogenomes and protein-coding genes of Chilopoda.

Species	Mitochondrial Genome			PCGs		
	A + T	AT Skew	GC Skew	A + T	AT Skew	GC Skew
*Bothropolys* sp.	70.6	0.07	−0.31	68.6	−0.13	−0.06
*C.s longicornis*	63.4	0.09	−0.32	60.5	−0.17	−0.05
*Lithobius forficatus*	67.9	0.09	−0.27	65.7	−0.14	−0.03
*M. marmoratus*	69.5	0.12	−0.33	67.1	−0.11	−0.02
*S. maritima*	64.1	0.22	−0.33	62.1	−0.12	−0.02
*S. dehaani*	74.1	0.13	−0.34	59.6	−0.12	−0.01
*S. mutilans*	78.8	0.05	−0.26	77.2	−0.12	−0.04
*S. subspinipes*	72.7	0.06	−0.33	71.6	−0.14	0
*Scolopendra morsitans*	72.8	0.02	−0.38	71.6	−0.14	−0.02
*Scolopocryptops* sp.	71.6	0.03	−0.31	70.4	−0.14	−0.01
*S. coleoptrata*	69.4	0.04	−0.31	68.3	−0.14	−0.05

## Data Availability

The newly sequenced mitogenomes were submitted to the GenBank database under the accession numbers of *M. marmoratus* (KX774322) and *S. subspinipes* (MN642577).

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
