# Peer review of "A Rearrangement of the Mitochondrial Genes of Centipedes (Arthropoda, Myriapoda) with a Phylogenetic Analysis"

_genes, 2022, doi:10.3390/genes13101787_

Round 1

Reviewer 1 Report

Reviewer #: -The manuscript describes the Rearrangement of mitochondrial genes of centipedes (Arthropoda, Myriapoda) and phylogenetic analysis. The author thinks that the limitation of taxon sampling and differences in results collected lead to phylogenetic relationships of the Myriapoda remaining contentious. The title selected for the article is a bit broader than the data presented in it. The techniques used in the manuscript are up to date and the experiments are well performed. However, I found several typos that need clarification to support and strengthen their conclusion.

Title: It can be improved. 

The introduction can be improvised with the latest updates.

It could have been better if you provide the graphical abstract.

Do you use any reference genes for this study? do you use positive and negative controls? Please provide a brief description and more details about the primers used.

Why do you select only these 6 pairs of primers? No details in the text?

In the case of discussion, please relate your finding, and conclusion with the latest supportive references. This will improve the quality and understanding of the manuscript.

Since I found some degree of difficulty in reading and understanding certain parts of the manuscript, I feel this manuscript needs correction in the case of materials and methods and discussion. I do think that the manuscript contains important issues, and interesting approaches, which can lead to evolutionary evidence related to the phylogenetic and evolutionary histories of the Myriapoda. Therefore, I consider this manuscript suitable for publication after the suggested minor revision in Agronomy.

Author Response

Dear Professor,

We would like to thank you for giving us constructive suggestions which would help us both in English and in depth improve the quality of the paper. Here we resubmit a new version of our manuscript with the title “Rearrangement of mitochondrial genes of centipedes (Arthropoda, Myriapoda) and phylogenetic analysis”, which has been modified according to your suggestions. we were also made to correct the mistakes and improve the English of the manuscript. We make all the changes in red in the revised manuscript.

The following is which I have made in response to your questions and suggestions on the item by item basis.

Suggestion 1: Title: It can be improved.

Answer: Thank you for your valuable suggestions. Although we have only sequenced and analyzed the mitochondrial genomes of two species of the Chilopoda, we not only did a simple analysis of the phylogenetic relationship of the four main groups of Myriapoda, but also explored the relationships among the four genera of the Chilopoda based on the mitochondrial genomic data. After careful thought and discussion, no more suitable title was concluded. If you still think the title is too broad, please point it out again and we will think more carefully to find a suitable title.

Suggestion 2: The introduction can be improvised with the latest updates.

Answer: Thank you for your suggestions. After checking and searching, we found that we did not update several recently published literatures into the manuscript in time. Now we have modified the introduction based on these several pieces of literature, If there is any omission, please let us know and we will make further modifications.

Suggestion 3: The introduction can be improvised with the latest updates.

It could have been better if you provide the graphical abstract.

Answer: This suggestion is very good. We did ignore the graphic abstract before. According to your suggestion, we have added the graphic abstract to the revised manuscript this time. Since this is the first time for us to make a graphic abstract, please tell us if there is anything wrong, and we will revise it in time

Suggestion 4: Do you use any reference genes for this study? do you use positive and negative controls? Please provide a brief description and more details about the primers used.

Answer: Dear reviewer, maybe our writing has caused your confusion. In fact, two new sequenced mitogenomes were obtained by PCR amplification and sanger sequencing. And the reliability of the sequences was verified by homologous sequence alignment within NCBI, and in the process of PCR amplification, we have used positive and negative controls, However, this step is rarely reflected in the writing process, and if it is needed, please point out again that we will add this part in the next round of revision. All the primers were valid reference primers that had been used by the published species. After using and screening, the six most suitable primer pairs were obtained. And we have added each PCR amplification products Length in Table 1. If more detailed primer information needs to be added to the manuscript, please kindly point out again.

Suggestion 5: Why do you select only these 6 pairs of primers? No details in the text?

Answer: Six pair primers were valid reference primers that had been used by the published species. After using and screening, and complete mitochondrial genome sequences can be obtained by these six pair primers. In the revised manuscript, we have explained the source of primers and the length of PCR products. If more detailed primer information needs to be added to the manuscript, please kindly point out again.

Suggestion 6: In the case of discussion, please relate your finding, and conclusion with the latest supportive references. This will improve the quality and understanding of the manuscript.

Answer: Thank you for your suggestions, which I also think is very necessary. In the revised manuscript, we have sorted out and supplemented the content discussed. If you feel that there is room for further improvement in the revised manuscript, please kindly propose it again, and we will revise it again in a targeted manner.

Thank you again for your hard work. If you find any mistakes in the revised manuscript, please kindly tell us, and we will timely modify it according to your opinions.

Best wishes,

Jiajia Wang

Reviewer 2 Report

First of all I would like to congratulate the authors for this MS.

I would like you to address some comments I have:

Lines 27 to 35. You left the instructions from the template. This shows that it seems you did not pay attention to it. If you read all the MS you should be aware of it. Please pay attention to these small details. Please delete all of those lines.

Line 40. 3,300 should be 3300 as in English the comma in numbers should be written only if they have 5 or more figures.

Line 78 and 79. In Hymenoptera the gene order depends on the taxa, for example in ants the ancestral cluster seems to be MIQ  (Ruiz-Mena et al., 2022) so the genes involved are I and M and not I and Q. 

Line 110. Please use italics in the species names

Methods: Could you please state which PCRs were long and which one were short?

Table 1. Regarding primers, how did you choose the primers that do not have reference? How did you design them? Please clarify.

3. Results. Please change all "AT" for "A+T" and "GC" for "G+C".

Figure 1. Please make it larger and probably use other color where they can be better read.

Figure 2a. Could you place the Diplopoda legend a bit more towards the right? 

Figure 3 and 4. Consider make it a bit larger as you have enough space bot better visualization. Even consider to use an horizontal page (if possible) including both

Congratulations again for the MS

Author Response

Dear Professor,

We would like to thank you for giving us constructive suggestions which would help us both in English and in depth improve the quality of the paper. Here we resubmit a new version of our manuscript with the title “Rearrangement of mitochondrial genes of centipedes (Arthropoda, Myriapoda) and phylogenetic analysis”, which has been modified according to your suggestions. we were also made to correct the mistakes and improve the English of the manuscript. We make all the changes in red in the revised manuscript.

The following is which I have made in response to your questions and suggestions on the item by item basis.

Suggestion 1: Lines 27 to 35. You left the instructions from the template. This shows that it seems you did not pay attention to it. If you read all the MS you should be aware of it. Please pay attention to these small details. Please delete all of those lines.

Answer: Thank you for pointing out this mistake, which is indeed caused by our negligence and we have been deleted in the revised manuscript.

Suggestion 2: Line 40. 3,300 should be 3300 as in English the comma in numbers should be written only if they have 5 or more figures.

Answer: We have corrected the mistake.

Suggestion 3: Line 78 and 79. In Hymenoptera the gene order depends on the taxa, for example in ants the ancestral cluster seems to be MIQ (Ruiz-Mena et al., 2022) so the genes involved are I and M and not I and Q.

Answer: After reading the literature you provided, we find that this sentence is indeed inaccurate, we think it would be better to write the following sentence: Most hymenopterans have the trnI, trnQ and/or trnM genes in different positions, such as the trnM-trnI-trnQ order is the most common in Formicidae [33-34]. I wonder if this is appropriate. Looking forward to your reply.

Suggestion 4: Line 110. Please use italics in the species names

Answer: We have corrected the mistake.

Suggestion 5: Methods: Could you please state which PCRs were long and which one were short?

Answer: Thank you for your suggestion, which is important information that we have ignored. In the revised manuscript, we have added each PCR amplification products Length in Table 1. I hope this modification has not caused you any confusion. If necessary, we can further modify it.

Suggestion 6: Table 1. Regarding primers, how did you choose the primers that do not have reference? How did you design them? Please clarify.

Answer: We apologize for any misunderstanding caused by this table. In fact, all primers were selected from reference articles. I think you may be confused by the following pair of primers: Dco3F + Dn4R, Dn4F + Hcob, DcobF + D12SR, and D12SF + Dco1R. In fact, they are not our designed, which all references to the validated available primers in “Dong, Y.; Sun, H.; Guo, H.; Pan, D.; Qian, C.; Hao, S.; Zhou, K. The complete mitochondrial genome of Pauropus longiramus (Myriapoda: Pauropoda): implications on early diversification of the myriapods revealed from comparative analysis. Gene 2012, 505, (1), 57–65.”

Suggestion 7: Results. Please change all "AT" for "A+T" and "GC" for "G+C".

Answer: We have corrected the mistake.

Suggestion 8: Figure 1. Please make it larger and probably use other color where they can be better read.

Answer: We have changed the figure 1 according to your suggestion, I hope you are not disappointed with the current color scheme.

Suggestion 9: Figure 2a. Could you place the Diplopoda legend a bit more towards the right?

Answer: Thank you for your suggestion, we have modified the pictures in the revised manuscript, and hope you are not disappointed.

Suggestion 10: Figure 3 and 4. Consider make it a bit larger as you have enough space bot better visualization. Even consider to use an horizontal page (if possible) including both

Answer: Thank you for your suggestion, we recreated Figure 3 and modified Figure 4 with a view to achieving a unified visualization, we have remade Figure 3 and revised Figure 4 in order to achieve the effect of unified visualization. If we misunderstand your suggestion, please suggest it again and we will further modify it.

Thank you again for your hard work. If there is any deficiency in the modification, please kindly point out again. We will try our best to modify it.

Best wishes,

Jiajia Wang

Reviewer 3 Report

Wang et al compted the mitogenomes of M. marmoratus and S. maritime, and built the phylogeny of Myriapoda. I think their results is are clearly presented and helpful for the field. 

I found the stroy is easy to follow. However, I found only a few typoes below. Other than that, I don't have much to add.

L27-L35 Delete.

L117 and L121 Toyobo and Takara are Japanese companies. 

L250 marks -> markers

L332 revealed -> suggested. I think revealed is too strong.

Author Response

Dear Professor,

We would like to thank you for giving us constructive suggestions which would help us both in English and in depth improve the quality of the paper. Here we resubmit a new version of our manuscript with the title “Rearrangement of mitochondrial genes of centipedes (Arthropoda, Myriapoda) and phylogenetic analysis”, which has been modified according to your suggestions. we were also made to correct the mistakes and improve the English of the manuscript. We make all the changes in red in the revised manuscript.

The following is which I have made in response to your questions and suggestions on the item by item basis.

Suggestion 1: L27-L35 Delete.

Answer: Thank you for pointing out this mistake, which is indeed caused by our negligence and we have been deleted in the revised manuscript.

Suggestion 2: L117 and L121 Toyobo and Takara are Japanese companies.

Answer: We really didn't notice this wrong writing, thank you for pointing out this mistake. We have made corrections in the revised manuscript.

Suggestion 3: L250 marks -> markers

Answer: We have corrected the mistake.

Suggestion 4: L332 revealed -> suggested. I think revealed is too strong.

Answer: thank you for your suggestion, after seeing your suggestion, we also thin that suggested is more suitable than revealed.

Thank you again for your hard work. If you find any mistakes in the revised manuscript, please kindly tell us, and we will timely modify it according to your opinions.

Best wishes,

Jiajia Wang